# Work-related COVID-19 transmission in six Asian countries/areas: A follow-up study

Fan-Yun Lan[1,2], Chih-Fu Wei[1], Yu-Tien Hsu[3], David C. Christiani[1], Stefanos N. Kales[1,4]*

1 Department of Environmental Health, Harvard University T.H. Chan School of Public Health, Boston, Massachusetts, United States of America, 2 Department of Occupational and Environmental Medicine, National Cheng Kung University Hospital, College of Medicine, National Cheng Kung University, Tainan, Taiwan, 3 Department of Social and Behavioral Science, Harvard University T.H. Chan School of Public Health, Boston, Massachusetts, United States of America, 4 Department of Occupational Medicine, Cambridge Health Alliance, Harvard Medical School, Cambridge, Massachusetts, United States of America

* skales@hsph.harvard.edu

**Data Availability Statement:** All relevant data are within the manuscript and its Supporting Information files.

**Funding:** The authors received no specific funding for this work.

## Abstract

### Objective

There is limited evidence of work-related transmission in the emerging coronaviral pandemic. We aimed to identify high-risk occupations for early coronavirus disease 2019 (COVID-19) local transmission.

### Methods

In this observational study, we extracted confirmed COVID-19 cases from governmental investigation reports in Hong Kong, Japan, Singapore, Taiwan, Thailand, and Vietnam. We followed each country/area for 40 days after its first locally transmitted case, and excluded all imported cases. We defined a possible work-related case as a worker with evidence of close contact with another confirmed case due to work, or an unknown contact history but likely to be infected in the working environment (e.g. an airport taxi driver). We calculated the case number for each occupation, and illustrated the temporal distribution of all possible work-related cases and healthcare worker (HCW) cases. The temporal distribution was further defined as early outbreak (the earliest 10 days of the following period) and late outbreak (11th to 40th days of the following period).

### Results

We identified 103 possible work-related cases (14.9%) among a total of 690 local transmissions. The five occupation groups with the most cases were healthcare workers (HCWs) (22%), drivers and transport workers (18%), services and sales workers (18%), cleaning and domestic workers (9%) and public safety workers (7%). Possible work-related transmission played a substantial role in early outbreak (47.7% of early cases). Occupations at risk varied from early outbreak (predominantly services and sales workers, drivers, construction laborers, and religious professionals) to late outbreak (predominantly HCWs, drivers, cleaning and domestic workers, police officers, and religious professionals).

**Competing interests:** The authors have declared that no competing interests exist.

## Conclusions

Work-related transmission is considerable in early COVID-19 outbreaks, and the elevated risk of infection was not limited to HCW. Implementing preventive/surveillance strategies for high-risk working populations is warranted.

## Introduction

Coronavirus disease 2019 (COVID-19) was declared by the World Health Organization (WHO) as a pandemic on March 11, 2020 and its local transmission has been reported in many countries [1]. The transmission pathways and the related risk factors are of vital interest in efforts to control the outbreak [2–4].

Work-related transmission is a crucial contributor to infectious disease outbreaks. The characteristics of SARS-CoV-2 virus and its transmission patterns could lead to high transmission rates among workers. For example, cases of COVID-19 largely present with mild or no symptoms [5]. Also, some studies have found similar transmissibility from asymptomatic and symptomatic patients [6–8]. These characteristics could lead to a higher probability of work-related transmission as people with mild symptoms could continue to work, travel or otherwise conduct business and spread the disease to others during work or commuting. Furthermore, the infected workers can subsequently transmit the virus to other people in their households and communities. Therefore, it is contingent to better understand the epidemiology of work-related transmission of COVID-19 to implement evidence-based prevention and protection strategies in workplaces.

Most of the studies to date focus on occupational exposure among healthcare workers (HCWs). Work-related transmission among HCWs constituted a large proportion in previous coronavirus outbreaks. HCWs comprised 37–63% of suspected severe acute respiratory syndrome (SARS) cases in highly affected Asian countries, and around 43.5% of Middle East respiratory syndrome (MERS) cases [9–11]. There was high prevalence of infection among HCWs despite the introduction of precautions against nosocomial transmission [12,13].

In contrast, there is limited discussion on the work-related risks among workers such as taxi drivers, tour guides, cleaners and janitors, and civil servants, who have frequent contact with the public in their daily routines or have workplaces with higher risks of virus exposure [14].

In this study, we aimed to identify the occupations at higher risk of COVID-19 transmission, and to explore the temporal distribution of work-related cases among local transmission.

## Materials and methods

### Study population selection

We extracted and included all locally transmitted COVID-19 confirmed cases from the publicized government investigation reports from six Asian countries/areas, including Hong Kong [15], Japan [16], Singapore [17], Taiwan [18], Thailand [19], and Vietnam [20]. These countries/areas were selected since they shared some common temporo-spatial characteristics. First, they are proximal to Mainland China, where the first outbreak of COVID-19 was reported. Second, the first cases of these countries/areas were imported cases from Mainland China in mid-January. Third, the first locally transmitted cases in these countries/areas were identified around late January to early February. We followed each country/area for forty days

since the report of the first locally transmitted case and excluded the imported cases. The study population selection process is presented in Fig 1.

## Categorization of work-related cases

The investigation reports of the six countries/areas mostly included case information such as age, sex, and brief contact/medical history. An example is: "*The new case was a 35 year-old female health care worker, and had close contacts with the X^th confirmed case* [19]." There were also sections indicating infection clusters: "*Four of the confirmed cases (Cases W, X, Y and Z) are linked to the XXX construction site* [17]." Occupational history was not always available. In most cases, if the contact history was obvious such as family cluster, the report would likely lack an occupational description. Based on the available information, each case report was reviewed by two occupational physicians and categorized for work-relatedness. Subsequently, the possible work-related cases were grouped into two categories based on whether the transmission source was known:

1. *Category 1*: *had clear close contact with a confirmed case due to work*, such as a registered nurse having a history of directly caring for a patient who is an index confirmed case; and

2. *Category 2*: *unknown transmission source; no apparent contact history but likely to be infected in the working environment*, such as an airport taxi driver without clear contact history to any confirmed case.

The cases with occupations and contact histories were coded according to the International Standard Classification of Occupations, 2008 (ISCO-08) [21]. We defined health professionals, medical doctors, and nursing professionals as healthcare workers (HCWs) regarding the high risk of infection due to the work. We further grouped the occupations into drivers and transport workers, services and sales workers, cleaning and domestic workers, public safety workers, religious workers, construction workers, and other groups according to the jobs similarity.

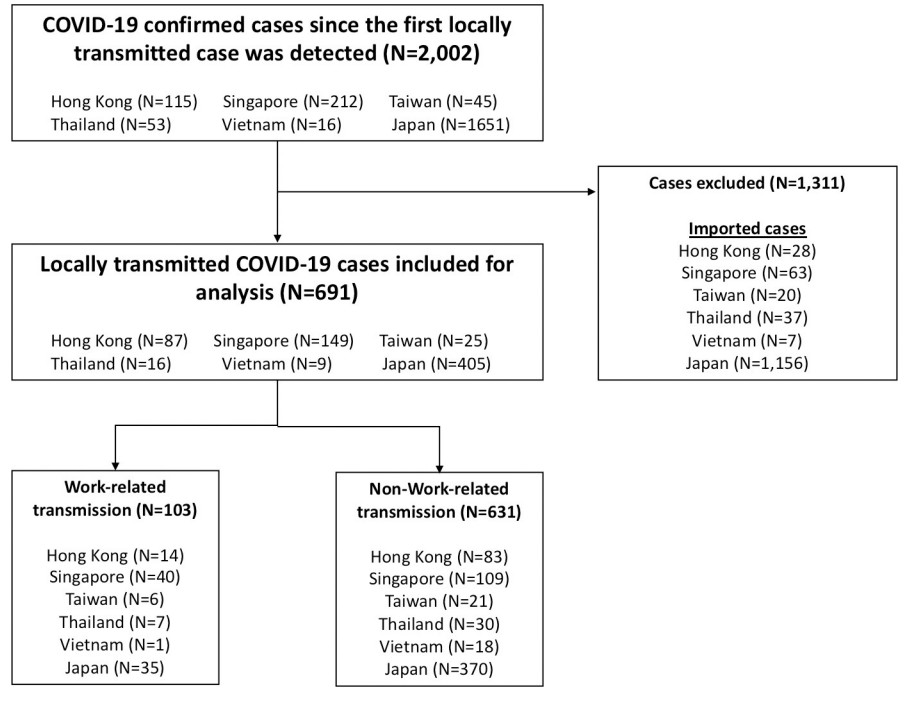

**Fig 1. Study population selection process.**

All differences between the occupation physicians were reviewed by the third investigator, who is a physician-epidemiologist to reach a consensus.

### Statistical analysis

Descriptive analysis was performed to compare the trends of daily reported cases in the locally transmitted cases, work-related cases, and HCW groups.

For each country/area, we calculated the intervals between the first reported locally transmitted case and the first reported work-related case, as well as the interval between the first reported locally transmitted case and the first reported HCW case. We also summarized the number of cases for each occupation across country/area and stratified the cases into early and late transmission periods. We defined early transmission period as the first 10 days from when the first locally transmitted case was reported, and late transmission period as the 11[th] to 40[th] day of the study period. We enlisted the most common occupations in each period and compared the distribution of occupations in order to examine the temporal difference. We performed Chi-squared tests or Fisher exact tests to compare the proportions of work-related cases and HCW cases among all local transmissions between early and late transmission periods.

We also conducted sensitivity analysis comparing the results between the six countries/areas and five countries/areas excluding Japan. We excluded Japan due to its different case reporting system from other countries/areas. Unlike other countries/areas that have central reporting systems providing cases' occupation and other contact history in a standardized form, Japan's reporting system is prefecture-based, where each prefecture reports separately, without consistent occupational coding. For example, some prefectures reported a case as a taxi driver; while some prefectures only reported a driver, without specifying the type of vehicle he/she drove [16,22,23]. Differences in reporting mechanisms and case information across prefectures could potentially bias the results. The descriptive analysis was performed by R software (version 3.6.2). The figures were plotted by Microsoft® **EXCEL**™.

## Results

We identified 2,002 officially confirmed COVID-19 cases within the designated 40-day interval among the six countries/areas. We excluded 1,312 imported cases and included 690 locally transmitted cases in the final analysis. The cases included in this study were reported between January 23, 2020 and March 14, 2020 (S1 Table).

103 possible work-related cases were included for analysis (including 37 cases from Category 1 and 66 from Category 2), accounting for 15% of local transmissions. Among the possible work-related cases, 22% were HCW. In addition to HCWs, we identified other occupations that were at higher risk of work-related transmission. The high-risk occupations included car, taxi and van drivers (N = 15), shop salesperson (N = 7), domestic housekeepers (N = 7), religious professionals (N = 6), construction laborers (N = 5), tour guides (N = 5), and so on. In terms of occupation grouping, the groups with the most cases were HCWs, drivers and transport workers, services and sales workers, cleaning and domestic workers, and public safety workers. (Table 1)

There were 31 incident confirmed cases during early transmission period, while there were 72 incident cases occurring in late transmission period. The most common occupations during early transmission were shop salesperson (N = 6), car, taxi and van drivers (N = 5), construction laborer (N = 4), religious professionals (N = 3), tour guides (N = 3), and receptionist (N = 3). Meanwhile, most common occupations during late transmission were health

**Table 1. Possible work-related COVID-19 cases categorized by occupation.**

| Occupation group | N (%) | Occupation (ISCO-08) | N (%) |
|---|---|---|---|
| Health professional (Healthcare workers) | 23 (22) | Other health professionals | 10 (10) |
| | | Nursing professionals | 10 (10) |
| | | Medical doctors | 3 (3) |
| Drivers and Transport workers | 19 (18) | Car, taxi and van drivers | 15 (15) |
| | | Locomotive engine drivers and related workers | 2 (2) |
| | | Bus and tram drivers | 2 (2) |
| Services and sales workers | 19 (18) | Shop salespersons | 7 (7) |
| | | Travel attendants, conductors and guides | 5 (5) |
| | | Receptionists | 3 (3) |
| | | Waiter or bartenders | 2 (2) |
| | | Cooks | 1 (1) |
| | | Personal care workers in health services | 1 (1) |
| Cleaning and domestic workers | 9 (9) | Domestic housekeepers | 7 (7) |
| | | Domestic cleaners and helpers | 2 (2) |
| Public safety workers | 7 (7) | Police officers | 3 (3) |
| | | Fire fighter | 2 (2) |
| | | Security guards | 2 (2) |
| Religious workers | 6 (6) | Religious professionals | 6 (6) |
| Construction workers | 5 (5) | Construction laborers | 5 (5) |
| Other groups | 15 (15) | Unspecified[a] | 15 (15) |
| Summary | | | 103 (100) |

ISCO-08: International Standard Classification of Occupations, 2008

[a]Mainly from workplace clusters without detailed occupational description of each case

professionals (N = 23), car, taxi and van drivers (N = 10), domestic housekeepers (N = 6), police officers (N = 3), and religious professionals (N = 3) (Table 2).

Notably, there were different composition of high-risk occupations across transmission periods. Car, taxi and van driver and religious professionals were the most common occupations in both early and late transmission periods. Retail salespersons and tour guides were

**Table 2. Occupation distribution of possible work-related COVID-19 cases in early and late transmission.**

| Occupation (ISCO-08) | Early transmission period, N = 31 | Late transmission period, N = 72 | P-value[a] |
|---|---|---|---|
| | Count, N (%) | Count, N (%) | |
| Health professionals | 0 (0) | 23 (32) | <0.001 |
| Shop salespersons | 6 (19) | 1 (1) | 0.003 |
| Car, taxi and van drivers | 5 (16) | 10 (14) | 0.767 |
| Domestic housekeepers | 1 (3) | 6 (8) | 0.672 |
| Construction laborers | 4 (13) | 1 (1) | 0.028 |
| Religious professionals | 3 (10) | 3 (4) | 0.362 |
| Police officers | 0 (0) | 3 (4) | 0.552 |
| Travel attendants, conductors and guides | 3 (10) | 2 (3) | 0.159 |
| Receptionists | 3 (10) | 0 (0) | 0.025 |

ISCO-08: International Standard Classification of Occupations, 2008

[a]P-values were calculated with Fisher exact test.

predominant in the early transmission period, while HCWs, domestic housekeepers, and police officers were the leading high-risk occupations in the late transmission period. Furthermore, while the proportion of work-related transmission decreased for shop salespersons, construction laborers and receptionists, there was a discernable increase in proportion of HCWs in the late period ($P <0.001$, Table 2)

Fig 2A illustrates new daily confirmed local transmission, possible work-related transmission, and HCW cases over time in the six countries/areas. While the number of daily confirmed local transmission increased, the number of work-related cases reported in each day remained relatively steady throughout the follow-up period. We found 48% of locally transmitted cases in the early transmission period were due to possible work-related transmission, compared to 11% in the late transmission period (Chi-squared statistic = 61.84, $P <0.0001$).

In further sensitivity analysis excluding Japan because of its different case reporting system, the daily confirmed local transmissions became relatively constant (Fig 2B). After excluding Japan, possible work-related cases comprised 44% of the locally transmitted cases in the early period, while only 18% in the late period (Chi-squared statistic = 18.8, $P$-value<0.0001).

HCW comprised 22% of the possible work-related cases. In the sensitivity analysis excluding Japan, the proportion of HCWs decreased to 7%. Moreover, we found the occurrence of

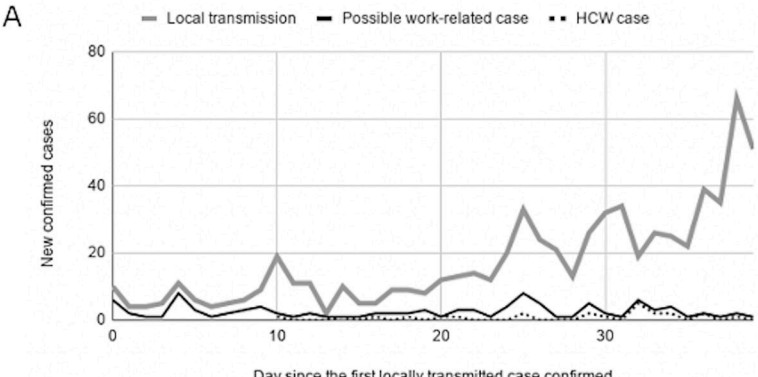

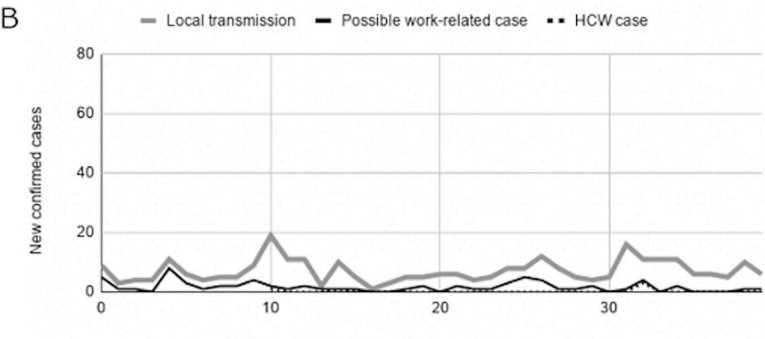

**Fig 2. New daily confirmed COVID-19 cases within 40-day follow-up periods across countries/areas.** (A) New daily confirmed Covid-19 cases within 40-day follow-up periods among the six countries/areas. (B) New daily confirmed Covid-19 cases within 40-day follow-up periods among the five countries/areas excluding Japan. HCW: Healthcare worker.

COVID-19 transmission among the HCW was relatively late compared to the non-HCW population. Fig 2A and Fig 2B showed a two-week lag of the first HCW case after the local COVID-19 outbreak (median lag: 15 days, IQR 13–20 days). The median time lag from the report of the first possible work-related case to the first HCW case was 13.5 days (IQR: 12.3–14.5 days) among the study population. In further sensitivity analysis excluding Japan, the median lags were 14 days (range: 10–32 days) and 13 days (range: 10–14 days), respectively.

Furthermore, nearly all the HCWs (95%) had clear and traceable contact history with a confirmed case (Category 1); while only 43.2% of the non-HCW cases could trace back the infection source (*P*-value<0.001).

## Discussion

In this study, we identified several high-risk occupations for COVID-19 infection that are rarely discussed [24]. These high-risk occupations comprised almost a half of local transmission during the early period of outbreak. In terms of the occupational risks of COVID-19 infection among the HCW, we found a median of two-week lag of HCW case after local transmission outbreaks. Moreover, non-HCW comprised the majority of the possible work-related cases and most of the cases were not able to trace back the infection sources.

Our results indicate the importance of work-related transmission in the local COVID-19 outbreak. One novel finding of this study is that the early transmissions were highly related to some occupations beyond healthcare settings, including taxi driver, salesperson, tour guide, and housekeeper and cleaner. Taxi drivers, salespersons and tour guides are at higher risk because of frequent contact with travelers. In fact, one of the earliest locally transmitted cases in Taiwan was a taxi driver who took a passenger returning from mainland China. This case led to a family cluster of COVID-19 with four more locally transmitted cases [18]. Another example was an infected worker involved in a reported cluster leading to three more local cases within a household in Singapore [4]. On the other hand, housekeepers and cleaners are more likely to be exposed to contaminated surfaces than direct contact with COVID-19 patients [25].

In this study, the proportion of HCWs among locally transmitted cases was smaller than non-HCWs in the included countries/areas, 3% versus 12% respectively. The first cases HCWs appeared much later than the first non-HCW cases in all the study countries. The lower rate of HCW and the occurrence time lags among HCWs likely reflects improved triage, screening and isolation of COVID-19 patients in the healthcare setting, as well as better personal protective equipment (PPE) and hygiene among HCWs once knowledge and experience with outbreaks increase [26–28]. Health professionals are more equipped with infection control knowledge and concepts, are more aware of self-hygiene and more informed regarding new outbreaks compared to non-HCWs [29].

This study raises the importance of protecting high-risk non-HCWs for several reasons. First, the work-related risks of respiratory infection, including COVID-19 infection, among the occupations are often neglected, and the workers are less likely to have PPEs or proper infection control in their workplaces. Second, it is much challenging to trace back the infection source of the non-HCW cases compared to the HCW cases, indicating the urging need of precautions for the high-risk population. Third, many of these occupations are impossible to work remotely and the workers may not benefit from the measures of worker-protection, such as government-imposed shutdown or work-from-home order. Fourth, many of the high-risk workers are in relatively lower socioeconomic status (SES), which is a risk factor of having COVID-19 infection and worse disease outcomes [30]. People from the lower ends of the society are more susceptible to infectious outbreaks due to poorer living and working conditions

[31,32]. They are more likely to have chronic health conditions which could lead to more severe consequences after being infected [33]. Protecting the high-risk workers could provide an opportunity to prevent the spread of the disease and to mitigate the deepening of health disparities.

The substantial contribution of non-HCW to the COVID-19 locally transmitted cases emphasizes the importance of implementing effective infection control in the non-healthcare workplaces to protect the workers in this pandemic [34]. Early delivery of infection control knowledge and health concepts to workers, as well as providing adequate PPE are crucial in protecting workers and the whole society.

Our study has some strengths. First, the data were extracted from the investigation reports published by the government of six countries/areas, which should be valid [35,36], and pooling of multi-county sources prevented the results from being skewed by single-country experience. Regarding the different case reporting system in Japan, we did further sensitivity analysis using the data from other five countries/areas and found similar temporal distribution patterns, which strengthened our conclusions. Second, every eligible case was reviewed by two occupational physicians and a physician epidemiologist with agreements on work-relatedness after thorough inspection of case reports. Moreover, we followed each country/area for 40 days, which allowed us to obtain comparable data for pooled analysis and illustrate trends of transmission in early stages of COVID-19 outbreak.

Nonetheless, there are limitations of this study. First, there were discrepancies in reporting and investigation across the countries/areas. Cases without reported occupational history could potentially lead to underestimation in the analysis. Second, the report date of a case could be different to the date of getting infected and having symptoms. However, the information bias should be non-differential as the official reports were not different between whether a case was work-related or not. Third, the criteria of deciding whom to be tested varied between countries/areas, especially during early outbreaks when testing capacities were limited. Therefore, high risk populations, including high risk occupations, might tend to be tested. However, we believe the bias was non-differential, as health authorities should not decide whom to be tested differently based on whether the suspected case was a worker or not. In fact, most of the early cases were tested because of the symptoms or obvious contact histories, instead of occupations [37]. Finally, we excluded all imported cases in the analysis. Travelers, however, could actually be business travelers, or other workers in travel-related industries, such as flight attendants, tour managers, and so on. Although workers of these occupations do have frequent contact with the public and have higher probability to be infected, our results could not demonstrate their risks and thus further studies on business travelers are warranted.

In conclusion, our study demonstrates that occupational infections are considerable in early COVID-19 local transmission. Second, several specific professional groups were at higher risk during early domestic outbreaks. We urge authorities to implement preventive strategies for each of these high-risk working populations.

## Supporting information

**S1 Table. Information of the 103 possible work-related COVID-19 cases during the study period.**
(DOCX)

## Author Contributions

**Conceptualization:** Fan-Yun Lan, Chih-Fu Wei, Yu-Tien Hsu, Stefanos N. Kales.

**Data curation:** Fan-Yun Lan, Chih-Fu Wei, Yu-Tien Hsu.

**Formal analysis:** Chih-Fu Wei, Yu-Tien Hsu.

**Funding acquisition:** David C. Christiani, Stefanos N. Kales.

**Investigation:** Fan-Yun Lan.

**Methodology:** Fan-Yun Lan, Chih-Fu Wei, Yu-Tien Hsu, David C. Christiani, Stefanos N. Kales.

**Project administration:** Stefanos N. Kales.

**Supervision:** David C. Christiani, Stefanos N. Kales.

**Validation:** David C. Christiani, Stefanos N. Kales.

**Writing – original draft:** Fan-Yun Lan.

**Writing – review & editing:** Chih-Fu Wei, Yu-Tien Hsu, David C. Christiani, Stefanos N. Kales.

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
