## [Decision Letter · Decision Letter 0]

27 Apr 2020

PONE-D-20-09603

Work-related Covid-19 transmission

PLOS ONE

Dear Dr. Kales,

Thank you for submitting your manuscript to PLOS ONE. After careful consideration, we feel that it has merit but does not fully meet PLOS ONE’s publication criteria as it currently stands. Therefore, we invite you to submit a revised version of the manuscript that addresses the points raised during the review process.

Please respond on a point-by-point basis to the reviewer comments and revise the manuscript accordingly.

We would appreciate receiving your revised manuscript by Jun 11 2020 11:59PM. To enhance the reproducibility of your results, we recommend that if applicable you deposit your laboratory protocols in protocols.io, where a protocol can be assigned its own identifier (DOI) such that it can be cited independently in the future. For instructions see: http://journals.plos.org/plosone/s/submission-guidelines#loc-laboratory-protocols

We look forward to receiving your revised manuscript.

Kind regards,

Jeffrey Shaman

Academic Editor

PLOS ONE

Journal Requirements:

2. Please consider modifying your title to ensure that it is specific, descriptive, concise, and comprehensible to readers outside the field (for example by specifying the nature of the study, and the names of the countries analysed ). When making changes please ensure that you amend the title on the online submission form (via Edit Submission) and the title in the manuscript so that they are identical.

3. In your Methods section, please detail the data sources used.

5. Please include a caption for figures 1 and 2.

6. Please include captions for your Supporting Information files at the end of your manuscript, and update any in-text citations to match accordingly. Please see our Supporting Information guidelines for more information: http://journals.plos.org/plosone/s/supporting-information

Reviewers' comments:

Reviewer's Responses to Questions

**Comments to the Author**

1. Is the manuscript technically sound, and do the data support the conclusions?

Reviewer #1: Yes

Reviewer #2: Yes

2. Has the statistical analysis been performed appropriately and rigorously? 

Reviewer #1: I Don't Know

Reviewer #2: N/A

3. Have the authors made all data underlying the findings in their manuscript fully available?

Reviewer #1: No

Reviewer #2: Yes

4. Is the manuscript presented in an intelligible fashion and written in standard English?

Reviewer #1: Yes

Reviewer #2: Yes

5. Review Comments to the Author

Reviewer #1: In this manuscript, the authors calculated the case number for each occupation, and illustrated the temporal distribution of all possible work-related cases and healthcare worker (HCW) cases by extracting confirmed COVID-19 cases from governmental investigation reports in six countries/areas. The results indicate the importance of work-related transmission in the local COVID-19 outbreak. They further found that the proportion of HCWs among locally transmitted cases was smaller than non-HCWs in the included countries/areas, emphasizing the importance of implementing effective infection control in the non-healthcare workplaces to protect the workers in this pandemic. This study provides a new insight in understanding the epidemiology of work-related transmission of COVID-19 and implement evidence-based prevention and protection strategies in workplaces. One concern from me is if there should be some difference among these countries and what should be the reasons?

Reviewer #2: The objective of this study was to describe the occupations among early COVID-19 cases arising from local transmission in five Asian countries for cases where occupational was thought to contribute to disease transmission, based on governmental reports. The investigators look at two time period, first week after introduction and the next month. Infections among healthcare workers appeared in the second period, and there were suggestions of some other changes in the distribution but interpretation is limited by the small sample size. It is important to understand what occupations put individuals at risk of COVID-19 to inform the need for intervention at the individual level and the society level.

Specific comments:

1. It would be helpful to know more about the data, including extraction of occupation and the judgement of occupationally-related disease transmission. For example, how frequently was occupation information missing? What types of occupations were identified that were judged not related to disease transmission? How was occupation recorded in the governmental reports, and did this vary among countries? What is the confidence in the role of occupation when an infection source could not be identified?

2. The sensitivity analysis with Japan is not well justified. Why was it thought that variation data recording practices among prefectures in Japan was greater among the four other coutnries? What was the evidence of this? The sensitivity results suggest that adding Japan wasn't a big impact on the results.

3. The results section text is very repetitive.

4. Figure 2 is difficult to read in black and white, or for people with color vision problems. Suggest using different line textures or thicknesses to enhance readability.

6. PLOS authors have the option to publish the peer review history of their article (what does this mean?). If published, this will include your full peer review and any attached files.

Reviewer #1: No

Reviewer #2: No

---

## [Author Response · Author response to Decision Letter 0]

29 Apr 2020

Journal Requirements:

Response: Thank you. We have revised the format/file names according to PLOS ONE’s requirements throughout the paper.

2. Please consider modifying your title to ensure that it is specific, descriptive, concise, and comprehensible to readers outside the field (for example by specifying the nature of the study, and the names of the countries analysed ). When making changes please ensure that you amend the title on the online submission form (via Edit Submission) and the title in the manuscript so that they are identical.

Response: Thank you. The paper title has been modified to “Work-related COVID-19 transmission in six Asian countries/areas: a follow-up study”, and the change has been made throughout the submission.

3. In your Methods section, please detail the data sources used.

Response: Thank you. The data sources have been added as references, as follows.

We extracted and included all locally transmitted COVID-19 confirmed cases from the publicized government investigation reports from six Asian countries/areas, including Hong Kong [15], Japan [16], Singapore [17], Taiwan [18], Thailand [19], and Vietnam [20].

15. Centre for Health Protection [Internet]. Media Room: Press Releases; c2020 [cited 2020 April 28]. Available from: https://www.chp.gov.hk/en/media/116/index.html

16. Ministry of Health, Labour and Welfare [Internet]. 報道発表資料; c2020 [cited 2020 April 28]. Available from: https://www.mhlw.go.jp/stf/houdou/index.html

17. Singapore Ministry of Health [Internet]. Past Updates on COVID-19 Local Situation; c2020 [cited 2020 April 28]. Available from: https://www.moh.gov.sg/covid-19/past-updates

18. Taiwan Centers for Disease Control [Internet]; c2020 [cited 2020 April 28]. Available from: https://www.cdc.gov.tw/En

19. Department of Disease Control [Internet]; Corona Virus Disease (COVID-19): Press Release; c2020 [cited 2020 April 28]. Available from: https://ddc.moph.go.th/viralpneumonia/eng/news.php

20. Ministry of Health [Internet]; c2020 [cited 2020 April 28]. Available from: https://ncov.moh.gov.vn

Response: Thank you. Dr Kales’ (the corresponding author) account has been linked to his ORCID iD. 

5. Please include a caption for figures 1 and 2.

Response: Thank you. The captions of Fig 1 and Fig 2 have been embedded in the manuscript body. 

6. Please include captions for your Supporting Information files at the end of your manuscript, and update any in-text citations to match accordingly. Please see our Supporting Information guidelines for more information: http://journals.plos.org/plosone/s/supporting-information

Response: Thank you. The caption of S1 Table has been included at the end of the manuscript (after References). The in-text citation has also been updated to match accordingly.

Reviewer #1: In this manuscript, the authors calculated the case number for each occupation, and illustrated the temporal distribution of all possible work-related cases and healthcare worker (HCW) cases by extracting confirmed COVID-19 cases from governmental investigation reports in six countries/areas. The results indicate the importance of work-related transmission in the local COVID-19 outbreak. They further found that the proportion of HCWs among locally transmitted cases was smaller than non-HCWs in the included countries/areas, emphasizing the importance of implementing effective infection control in the non-healthcare workplaces to protect the workers in this pandemic. This study provides a new insight in understanding the epidemiology of work-related transmission of COVID-19 and implement evidence-based prevention and protection strategies in workplaces. One concern from me is if there should be some difference among these countries and what should be the reasons?

Response: Thank you. Different countries do have different disease investigation and reporting policies, probably due to various political systems, historical factors, religious reasons, and so on. In the revised discussion, we addressed the limitation that there were discrepancies in reporting and investigation across the countries/areas. This is also the reason we excluded Japan as a sensitivity analysis, because unlike other countries/areas that have central reporting systems, Japan reports cases from each prefecture separately. 

Reviewer #2: The objective of this study was to describe the occupations among early COVID-19 cases arising from local transmission in five Asian countries for cases where occupational was thought to contribute to disease transmission, based on governmental reports. The investigators look at two time period, first week after introduction and the next month. Infections among healthcare workers appeared in the second period, and there were suggestions of some other changes in the distribution but interpretation is limited by the small sample size. It is important to understand what occupations put individuals at risk of COVID-19 to inform the need for intervention at the individual level and the society level.

Specific comments:

1. It would be helpful to know more about the data, including extraction of occupation and the judgment of occupationally-related disease transmission. For example, how frequently was occupation information missing? What types of occupations were identified that were judged not related to disease transmission? How was occupation recorded in the governmental reports, and did this vary among countries? What is the confidence in the role of occupation when an infection source could not be identified?

Response: Thank you for your comments. Please find the point-by-point responses below.

1) We have added more detailed descriptions of the government reports, as follows.

The investigation reports of the six countries/areas mostly included case information such as age, sex, and brief contact/medical history. An example is: “The new case was a 35 year-old female health care worker, and had close contacts with the Xth confirmed case [19].” There were also sections indicating infection clusters: “Four of the confirmed cases (Cases W, X, Y and Z) are linked to the XXX construction site [17].” Occupational history was not always available. In most cases, if the contact history was obvious such as family cluster, the report would likely lack an occupational description. Based on the available information, each case report was reviewed by two occupational physicians and categorized for work-relatedness. 

2) Most cases judged not related to work were because of obvious non-occupational contact history or a lack of occupational history. Therefore, we did not determine particular types of occupations that were not related to disease transmission.

3) Different countries do have different disease investigation and reporting policies. Occupational history was not generally reported for each case. However, for the reports with available and adequate occupation information, we were able to determine the possible work-relatedness. We addressed this issue in the discussion as a limitation that cases without reported occupational history could potentially lead to underestimation in the analysis.

4) We were more confident in determining the work-relatedness if a case had close contact with a confirmed case due to work, or Category 1, such as a registered nurse having a history of directly caring for a patient who is an index confirmed case. As to Category 2, unknown transmission source; no apparent contact history but likely to be infected in the working environment, such as an airport taxi driver without clear contact history to any confirmed case, we had less confidence in work-relatedness but judged such cases to be probable (or more likely than not) work-related. 

2. The sensitivity analysis with Japan is not well justified. Why was it thought that variation data recording practices among prefectures in Japan was greater among the four other countries? What was the evidence of this? The sensitivity results suggest that adding Japan wasn't a big impact on the results.

Response: Thank you for your comments. We have revised the following sentences and added references accordingly as follows.

Unlike other countries/areas that have central reporting systems providing cases’ occupation and other contact history in a standardized form, Japan’s reporting system is prefecture-based, where each prefecture reports separately, without consistent occupational coding. For example, some prefectures reported a case as a taxi driver; while some prefectures only reported a driver, without specifying the type of vehicle he/she drove [16,22,23]. Differences in reporting mechanisms and case information across prefectures could potentially bias the results.

16. Ministry of Health, Labour and Welfare [Internet]. 報道発表資料; c2020 [cited 2020 April 28]. Available from: https://www.mhlw.go.jp/stf/houdou/index.html

22. Ministry of Health, Labour and Welfare [Internet]. 新型コロナウイルス感染症患者の発生について(第 2 報); c2020 [cited 2020 April 28]. Available from: https://www.mhlw.go.jp/content/10906000/000598149.pdf?fbclid=IwAR0SZCxLB4VruarnnbNNcsfyGG-Bb96y_KXFBIX3UR4t7B_BzhlbIA4_6Gk

23. Ministry of Health, Labour and Welfare [Internet]. 新型コロナウイルス感染症患者の発生について(県内8・9・10例目); c2020 [cited 2020 April 28]. Available from: https://www.mhlw.go.jp/content/10906000/000609268.pdf?fbclid=IwAR1bImIVvBWmrftmd0X0cCrmmg6HJdnD2F8UsXm3c9j-jtAEO3Hs8y0nYGA

Therefore, the exclusion of Japan is because of the concerns of potential misclassification. The most substantial impact after excluding Japan was the proportion of non-HCWs among work-related cases raised from 78% to 93% (shown in the revised Results).

3. The results section text is very repetitive.

Response: Thank you for your comments. We have streamlined the text of Results by incorporating sensitivity analysis results into the main results, as follow.

HCW comprised 22% of the possible work-related cases. In the sensitivity analysis excluding Japan, the proportion of HCWs decreased to 7%. Moreover, we found the occurrence of COVID-19 transmission among the HCW was relatively late compared to the non-HCW population. Fig 2A and Fig 2B showed a two-week lag of the first HCW case after the local COVID-19 outbreak (median lag: 15 days, IQR 13-20 days). The median time lag from the report of the first possible work-related case to the first HCW case was 13.5 days (IQR: 12.3-14.5 days) among the study population. In further sensitivity analysis excluding Japan, the median lags were 14 days (range: 10-32 days) and 13 days (range: 10-14 days), respectively. 

4. Figure 2 is difficult to read in black and white, or for people with color vision problems. Suggest using different line textures or thicknesses to enhance readability.

Response: Thank you for your comments. We have revised the figures.

---

## [Decision Letter · Decision Letter 1]

11 May 2020

Work-related COVID-19 transmission in six Asian countries/areas: a follow-up study

PONE-D-20-09603R1

Dear Dr. Kales,

We are pleased to inform you that your manuscript has been judged scientifically suitable for publication and will be formally accepted for publication once it complies with all outstanding technical requirements.

With kind regards,

Jeffrey Shaman

Academic Editor

PLOS ONE

Additional Editor Comments (optional):

Reviewers' comments:

Reviewer's Responses to Questions

**Comments to the Author**

1. If the authors have adequately addressed your comments raised in a previous round of review and you feel that this manuscript is now acceptable for publication, you may indicate that here to bypass the “Comments to the Author” section, enter your conflict of interest statement in the “Confidential to Editor” section, and submit your "Accept" recommendation.

Reviewer #1: All comments have been addressed

Reviewer #2: All comments have been addressed

2. Is the manuscript technically sound, and do the data support the conclusions?

Reviewer #1: (No Response)

Reviewer #2: Yes

3. Has the statistical analysis been performed appropriately and rigorously? 

Reviewer #1: (No Response)

Reviewer #2: Yes

4. Have the authors made all data underlying the findings in their manuscript fully available?

Reviewer #1: (No Response)

Reviewer #2: Yes

5. Is the manuscript presented in an intelligible fashion and written in standard English?

Reviewer #1: (No Response)

Reviewer #2: Yes

6. Review Comments to the Author

Reviewer #1: (No Response)

Reviewer #2: The authors have addressed the comments from this reviewer, and the manuscript is improved. The figures look better.

7. PLOS authors have the option to publish the peer review history of their article (what does this mean?). If published, this will include your full peer review and any attached files.

Reviewer #1: No

Reviewer #2: No

---

## [Editor Report · Acceptance letter]

12 May 2020

PONE-D-20-09603R1 

Work-related COVID-19 transmission in six Asian countries/areas: a follow-up study 

Dear Dr. Kales:

I am pleased to inform you that your manuscript has been deemed suitable for publication in PLOS ONE. Congratulations! Your manuscript is now with our production department. 

With kind regards,

on behalf of

Prof. Jeffrey Shaman 

Academic Editor

PLOS ONE